

# Biogeographic barriers drive co-diversification within associated eukaryotes of the *Sarracenia alata* pitcher plant system

Jordan D. Satler[1], Amanda J. Zellmer[2] and Bryan C. Carstens[1]

[1] Department of Evolution, Ecology and Organismal Biology, The Ohio State University, Columbus, OH, United States
[2] Department of Biology, Occidental College, Los Angeles, CA, United States

## ABSTRACT

Understanding if the members of an ecological community have co-diversified is a central concern of evolutionary biology, as co-diversification suggests prolonged association and possible coevolution. By sampling associated species from an ecosystem, researchers can better understand how abiotic and biotic factors influence diversification in a region. In particular, studies of co-distributed species that interact ecologically can allow us to disentangle the effect of how historical processes have helped shape community level structure and interactions. Here we investigate the *Sarracenia alata* pitcher plant system, an ecological community where many species from disparate taxonomic groups live inside the fluid-filled pitcher leaves. Direct sequencing of the eukaryotes present in the pitcher plant fluid enables us to better understand how a host plant can shape and contribute to the genetic structure of its associated inquilines, and to ask whether genetic variation in the taxa are structured in a similar manner to the host plant. We used 454 amplicon-based metagenomics to demonstrate that the pattern of genetic diversity in many, but not all, of the eukaryotic community is similar to that of *S. alata*, providing evidence that associated eukaryotes share an evolutionary history with the host pitcher plant. Our work provides further evidence that a host plant can influence the evolution of its associated commensals.

## INTRODUCTION

Dynamic processes during the Pleistocene epoch have been implicated as drivers of biological diversification (e.g., *Hewitt, 2000*; *Hewitt, 2004*). Glacial cycles contributed to both landscape changes and climatic oscillations, providing strong abiotic factors that have led to speciation within many groups (e.g., *Leaché & Fujita, 2010*; *McCormack et al., 2011*). One region strongly influenced by these processes during the Quaternary is the southeastern United States, where decades of research has examined the structure of genetic variation in a diverse set of taxa (e.g., *Avise et al., 1987*; *Avise, 2000*; *Burbrink, Lawson & Slowinski, 2000*; *Weisrock & Janzen, 2000*; *Jackson & Austin, 2010*; *Newman & Rissler, 2011*). Although glaciers never extended to this latitude, changes in both flow

Corresponding author
Bryan C. Carstens,
carstens.12@osu.edu

rate and direction of flow of major rivers coupled with fluctuations in sea level influenced phylogeographic patterns in this region (reviewed in *Soltis et al. 2006*). Specifically, major rivers in the region have produced population genetic structure in many clades, with the Mississippi River recognized as a well-characterized biogeographic barrier (*Brant & Orti, 2003*; *Pyron & Burbrink, 2009*). The influence of landscape features coupled with the presence of large-scale barriers can be expected to isolate populations within a species, especially those with limited dispersal abilities. Consequently, plants and animals that lack the ability to traverse large bodies of water are expected to exhibit substantial population genetic structure in this region.

Complex interactions that occur within ecological communities can influence the formation and maintenance of biodiversity. For example, numerous studies have shown how host plant diversification can contribute to the diversification of associated species, typically insects (e.g., *Farrell & Mitter, 1990*; *Wheat et al., 2007*; *McKenna et al., 2009*; *Espíndola, Carstens & Alvarez, 2014*). These include systems where plants evolve secondary compounds in an "escape and radiate" model of coevolution (*Ehrlich & Raven, 1964*), and systems that include mutualist organisms such as plants and their pollinators. Such interactions can result in congruent demographic histories (e.g., *Smith et al., 2011*) and patterns of co-diversification (e.g., *Rønsted et al., 2005*). While it seems clear that the ecological interactions among plants and associated arthropods (e.g., herbivores and pollinators) can potentially drive patterns of co-diversification, it is unclear how host plants may influence other commensal organisms, particularly small eukaryotes. Communities of commensal organisms in both facultative and obligate relationships may be expected to show varying evolutionary patterns attributed to the level of dependency on the host plant. Given the dynamic and topologically complex landscape of the southeastern region, the study of ecological communities that span the breadth of host affinity, dispersal ability and life history traits can help inform how taxonomically diverse communities have assembled through time, and whether present day ecological associations extend into the deep past.

Phytotelmata—water bodies contained within living plants—provide an ideal system for investigating co-diversification within an ecological community because they are self contained and discrete units (*Kitching, 2000*). Carnivorous pitcher plants are one such system, where decades of ecological work have documented a complex and distinct ecosystem associated with the pitcher fluid contained within the modified leaves. Pitcher plants in the genus *Sarracenia* (F. Sarraceniaeceae) contain a diverse microbiome, including groups such as bacteria, algae, protists, rotifers and arthropods (e.g., *Folkerts, 1999*; *Miller & Kneitel, 2005*; *Peterson et al., 2008*; *Koopman et al., 2010*). Their highly modified leaves form a trap that captures and digests prey items, while also providing a unique habitat for commensal organisms. Associated inquilines form complex relationships in the pitchers, with many supplying digestive enzymes that help break down decomposing prey items providing inorganic compounds for the plant (see *Adlassnig, Peroutka & Lendl, 2011*). A wide range of ecological work has investigated the communities associated with these plants, primarily in *Sarracenia purpurea*, showing community structure and interactions among the inquilines (e.g., *Addicott, 1974*; *Bradshaw & Creelman, 1984*; *Buckley et al., 2003*; *Gotelli & Ellison, 2006*; *TerHorst, Miller & Levitan, 2010*; *Miller & TerHorst, 2012*). Here, we focus

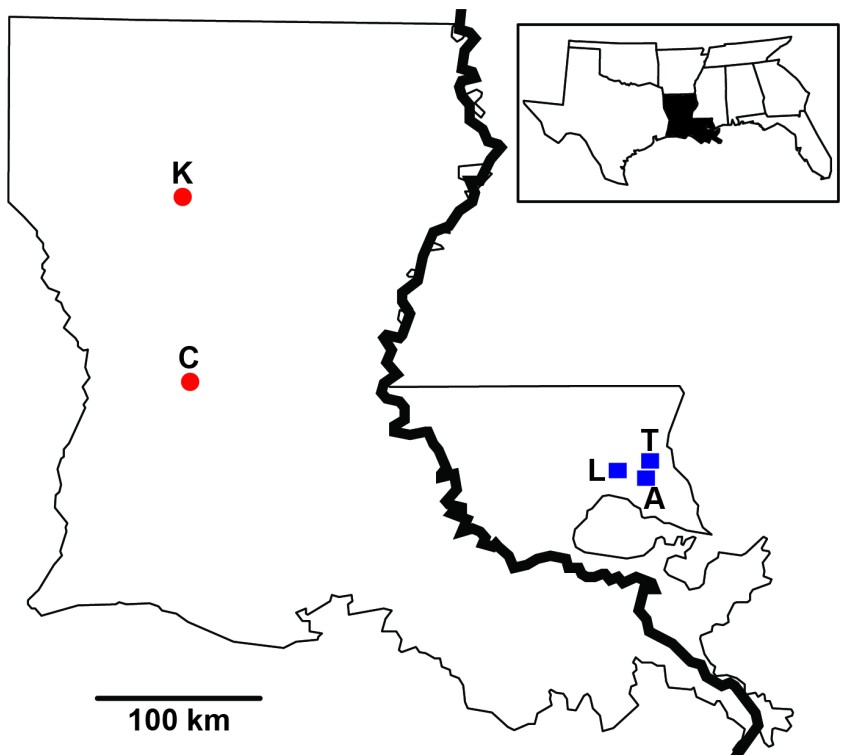

**Figure 1** **Sampling distribution of *Sarracenia alata* in Louisiana.** Sample sites are partitioned based on side of the Mississippi River. Red circles represent Kisatchie (K) and Cooter's Bog (C) in the west; blue squares represent Lake Ramsey (L), Abita Springs (A) and Talisheek (T) in the east.

on the Pale Pitcher Plant *Sarracenia alata*, a species distributed in patchy habitats along the gulf coast across eastern Texas, Louisiana, Mississippi and Alabama. This species is largely isolated from its congeners and occupies disjunct eastern and western regions across the Mississippi River (Fig. 1). Work by *Koopman & Carstens (2010)* identified population genetic structure in *S. alata*, and *Zellmer et al. (2012)* showed that major rivers in the region promoted diversification within the plant. Population divergence across either side of the Mississippi River is likely well into the Pleistocene, and estimated at greater than 120,000 years before present (*Zellmer et al., 2012*). Further analysis suggests that *S. alata* may contain two cryptic species, corresponding to populations on the eastern and western sides of the Mississippi River (*Carstens & Satler, 2013*). *Sarracenia alata* thus represents a particularly attractive system for investigating patterns of co-diversification, because the species exhibits strong genetic differentiation across the landscape, with significant divergence across an important biogeographic barrier (*Soltis et al., 2006*). In addition, longleaf pine savannahs in the south have seen a staggering amount of habitat loss in recent times (∼1% of its original habitat remains; *Noss, 1989*). High levels of cryptic genetic diversity highlight *S. alata* as a species of interest; identifying ecologically associated taxa with a shared evolutionary history has clear conservation implications.

Phylogeographic investigations of co-distributed taxa are usually limited to particular taxonomic groups (e.g., *Bell et al., 2012*; *Fouquet et al., 2012*; *Smith, Amei & Klicka, 2012*;

*Hope et al., 2014*). While these studies can reveal evolutionary processes that produce patterns within biogeographic regions, the conclusions drawn from such findings can be limited by the shared life history traits that influence the formation of genetic structure (e.g., dispersal ability, population size). Metagenomics provides a powerful approach for efficiently and rapidly sampling taxonomic diversity within a habitat (reviewed in *Tringe & Rubin, 2005*), and thus may provide comparative phylogeographic investigations with an efficient approach to the sampling of taxa. Through the sequencing of environmental DNA, communities of small to microscopic organisms can be directly sampled from the environment resulting in the assemblage of a data set spanning a wide taxonomic breadth. Thus, when coupled with next generation sequencing methods (*Mardis, 2008*), metagenomics greatly increases the "taxonomic toolbox" lending itself well to investigations of comparative phylogeography. By analyzing a disparate assemblage of taxa comprising an ecological community, our work has the potential to reveal a shared response to historical events and thus evidence that evolutionary processes can shape community structure and interactions through time (*Smith et al., 2011*). With the diverse array of microscopic inquilines present within *Sarracenia* (e.g., *Miller & Kneitel, 2005*), pitcher plants provide an ideal system for understanding how a host plant may influence genetic variation within an associated community, and metagenomics provides a tool for sampling this taxonomic diversity.

Here we explore the process of evolutionary diversification in an ecological community. We directly sample pitcher fluid from the modified leaves of *S. alata*, and apply a novel approach utilizing metagenomics to test if *S. alata* has influenced genetic structure in its eukaryotic commensal organisms. First, we characterize taxonomic diversity within the pitcher plant fluid to get an understanding of the major lineages and their abundance in this unique habitat. We then generate a comparative data set of OTUs which span the Mississippi River, and assess the degree to which the inquiline community shares population genetic structure with the host plant. We hypothesize that if eukaryotes associated with *S. alata* are ecologically dependent on the plant, then the evolutionary history of the commensals should exhibit population genetic structure largely congruent with that of *S. alata*. Alternatively, if taxa do not share an evolutionary history with *S. alata*, community members should have unique population genetic structure indicating an idiosyncratic response to landscape processes driving diversification in the region.

## MATERIAL AND METHODS

### Genetic sampling

Pitcher fluid samples were collected following *Koopman & Carstens (2010)* during the spring and summer of 2009 from part of the plant's distribution. Specifically, samples were collected from 40 individuals across four locales (Abita Springs, Cooter's Bog, Kisatchie, Talisheek; Fig. 1) in Louisiana during June and August, as pitcher diversity peaks at this time (*Koopman et al., 2010*). In addition, fluid was collected from ten individuals per month for five months (April through August) from Lake Ramsey, resulting in 50 samples, for a total sampling effort of 90 individuals from five locales. Sampling was

originally designed to investigate both spatial (all five locales) and temporal (Lake Ramsey) dynamics, however, we focus on just spatial patterns in this study. DNA was extracted using the Powersoil DNA Isolation Kit (MO Bio, Carlsbad, CA, USA). The large subunit 28S rRNA region was amplified for each fluid sample using the following primer combination (LS1F: GTACCCGCTGAACTTAAGC ; LS4R: TTGTTCGCTATCGGTCTC; modified from *Hausner, Reid & Klassen, 1993*), targeting a roughly 330 base pair (bp) region. Each pitcher fluid sample was labeled with MID tags to allow for multiplexing of individuals. PCRs were performed in triplicate and then pooled to prevent PCR bias, and subsequently sequenced on a 454 Life Sciences Genome Sequencer FLX (Roche, Basel, Switzerland) at the Engencore Genomics Facility (University of South Carolina, Columbia) using 1/8th of a plate. Raw sequences were initially processed using Mothur (*Schloss et al., 2009*) to sort sequences by individual, remove low quality reads, and identify unique sequences for each individual. Chloroplast data for *S. alata* was gathered from a previous study (see *Carstens & Satler (2013)* for details).

## Bioinformatics

To quantify the taxonomic diversity present within the pitchers, sequences were clustered into operational taxonomic units (OTUs) through *de novo* assembly. Metagenomic studies commonly use *de novo* assembly for generating OTUs (e.g., *O'Brien et al., 2005*; *Bik et al., 2012*; *Zimmerman & Vitousek, 2012*), and this allowed for a rough characterization of the number of taxonomic units present within the pitcher fluid.

Sequences were first combined within each of the sampling locales (i.e., we restricted clustering to those sequences collected from within each sample site), thereby treating each of the five sites as a separate population (see Fig. 1). Reads were trimmed to 275 bp, discarding any sequences below this threshold to remove potential bias associated with clustering samples of unequal sequence size. For consistency, we only analyzed sequences from Lake Ramsey collected at the same time periods as from the other sampling sites. Trimmed sequences were assembled into clusters using the UPARSE algorithm (*Edgar, 2013*); this pipeline been shown to outperform commonly used clustering methods such as Mothur and QIIME, and to work well under a solely *de novo* clustering approach. Within each locale, identical reads were collapsed and abundance values recorded (i.e., the number of times each unique read appeared in the data set). Sequences were then clustered based upon a 97% threshold, with the most represented sequences (based on abundance values) used to form initial OTU clusters, using a dynamic programming algorithm to find clusters with the maximum score. The percent similarity threshold is subjective, but since it is required for *de novo* assembly, we justify this value by noting that (i) it was recommended by the author for *de novo* assembly in UPARSE (*Edgar, 2013*), (ii) it falls within the range used to delimit fungi with this locus (see *Sota et al., (2014)* and references within), a group expected to be well represented within the pitcher fluid, and (iii) chimeric detection is increasingly difficult when this value is decreased. The clustering step in the pipeline ("cluster_otus") uses UPARSE-OTU, an algorithm that simultaneously determines the OTU clusters while removing chimeric sequences from the data set, a potential problem due to errors with pyrosequencing.

Following OTU clustering, a single sequence from each cluster was used with a Basic Local Alignment Search Tool (BLAST) search to gather taxonomic identification for each of the clusters. Although there is a concern with the incompleteness of public databases, and that searches could return spurious matches (*Koski & Golding, 2001*; *Tringe & Rubin, 2005*), at a higher taxonomic level (e.g., Class, Order), we can be reasonably confident that sequence matches reveal organismal affinity. In addition, for the purposes of this study, a qualitative assessment of higher level identification is sufficient to understand taxonomic diversity present within the pitcher fluid. A custom python script was used to search for taxonomic identities among the OTUs. For all BLAST searches, sequences representing the centroid of the original OTU searches (in UPARSE) were queried against the NCBI nucleotide database, using the Megablast search algorithm, saving the top hit from each search. OTUs with BLAST hits were grouped by higher-level identification, generally at the level of Class or Order, to identify the variety of organisms present within the pitcher fluid. After summarizing taxonomic identity within the pitcher fluid at each site, rarefaction curves were generated with the package vegan (*Oksanen et al., 2015*) in R (*R Core Team, 2015*), as a means to test if the taxonomic diversity had been adequately captured with our sampling efforts.

## Population structure

The major goal of this study is to identify OTUs that span the Mississippi River, and test if the landscape processes that have influenced diversification in *S. alata* have influenced the sampled organisms in a similar manner. To generate a comparative data set, all raw sequences were combined and OTUs were assembled with UPARSE following the steps outlined above (i.e., all sequences were clustered in a global analysis, regardless of sampling location). This data set included all sequences generated from Lake Ramsey, as we were interested in collecting taxa with widespread distributions. If taxa were time dependent, they would be restricted to Lake Ramsey (during the months when only this locality was sampled) and removed following our filtering process (see below); however, taxa stable in these communities would comprise additional sequence information for comparative analysis. Following initial OTU clustering, the data set was reduced to those taxa that contained at least ten sequences per OTU with a minimum of three sequences on either side of the Mississippi River. These thresholds were used to maximize the number of OTUs represented in the final data set while still containing enough sequence data for statistical inference, both within and across sampling sites. In addition, it is expected that any potential chimeric sequences not removed in the clustering step will fall below these thresholds, further reducing the potential for error with our final OTUs. Each OTU was aligned with MAFFT (*Katoh & Standley, 2013*), using either the L-INS-i (<200 sequences) or FFT-NS-i (>200 sequences) algorithm. To survey taxonomic diversity among the retained OTUs, a BLAST search was conducted on each of the OTUs following the same steps as outlined above.

Data were summarized within each of the OTUs in order to characterize genetic variation and quantify population genetic structure. Standard population genetic summary statistics (nucleotide diversity ($\pi$), Watterson's theta ($\Theta_w$), and Tajima's $D$) were calculated with

the package Pegas (*Paradis, 2010*) in R. Several approaches were used to explore the partitioning of genetic variation among the OTUs. $G_{ST}$ (*Nei, 1973*) values were generated to estimate the degree of population differentiation among the locales, and were calculated with the R package gstudio (*Dyer, 2012*). The level of genetic partitioning was assessed with an analysis of molecular variation (AMOVA; *Excoffier, Smouse & Quattro, 1992*), because the $G_{ST}$ is an analog to $F_{ST}$ values (*Nei, 1973*). AMOVAs take into account the amount of variation in the sequence data, thereby extracting more information to determine the level of spatial structuring within the taxa. AMOVAs were calculated in the program SPADS (*Dellicour & Mardulyn, 2014*), with 10,000 permutations to generate levels of significance. Hierarchical levels tested included (i) sampling locales within each region (i.e., side of the Mississippi River), (ii) sampling locales within total distribution, and (iii) between regions. In addition, the amount of allelic sorting on either side of the Mississippi River was calculated using the genealogical sorting index (GSI; *Cummings, Neel & Shaw, 2008*). This method is commonly applied to tests of taxonomic distinctness; it is applied here to quantify levels of lineage sorting within each side of the river, with higher levels of sorting suggesting greater population genetic structure indicative of a longer period of population isolation. GSI values range from 0 (no sorting) to 1 (monophyletic on either size of barrier), with *p*-values indicating the extent to which genetic structure recovered is more than would be expected by chance alone. An input genealogy is required to calculate the GSI; these were estimated using Maximum Likelihood (ML) with RAxML v7.2.8 (*Stamatakis, 2006*; *Stamatakis, Hoover & Rougemont, 2008*). Depending on the number of sequences in the OTU, models of sequence evolution included either GTRCAT (>200 sequences) or GTRGAMMA (<200 sequences). Each ML tree was then input to the GSI web server, with 10,000 permutations to generate levels of significance. In addition, isolation by distance (IBD) values were calculated to see if there was a correlation between genetic and geographic distance, using the IBDWS v3.23 web server (*Jensen, Bohonak & Kelley, 2005*). Genetic distance matrices were calculated using a Kimura 2-parameter (K2P) substitution model for each OTU; geographic matrices were constructed measuring the Euclidean distance between sampling locales in kilometers with the distance measurement tool in Google Earth (www.google.com/earth/, last accessed 18 July 2015). Finally, we used a chi-squared goodness-of-fit test to see if the number of OTUs with significant population genetic structure across the various analyses was more than would be expected by chance alone (assuming $\alpha = 0.05$). This allowed us to test the null hypothesis that there is no correlation of population structure between the members of the eukaryotic community and the host plant.

## RESULTS

### Genetic sampling

High-throughput sequencing resulted in a total of 26,399 sequences across 90 sampled pitchers. Following demultiplexing and quality control of samples, an average of 101 unique sequences were retained per pitcher (range: 12–199) for a total of 9,045 sequences. A FASTA file containing all 9,045 sequences, as well as all OTU matrices from the comparative data set (see below), has been deposited at Dryad (doi: 10.5061/dryad.j3n0g).

## Taxonomic diversity

To remove biases associated with the clustering of length variable sequences, all sequences were trimmed to 275 bp (discarding any reads below this threshold), reducing the data set from 9,045 sequences to 8,991 sequences. Lake Ramsey contained a disproportionately larger percentage of the total number of sequences (49%); however, to compare samples collected from the same time periods, we only analyzed those samples from June and August, reducing the number of sequences from Lake Ramsey from 4,398 to 2,286, resulting in a total of 6,879 sequences. OTU clustering at the 97% sequence identity within each locale resulted in a median of 66 OTUs per sample site (324 total), ranging from 48 (Cooter's Bog) to 82 (Lake Ramsey) total OTUs, with an average of 21 sequences per OTU when averaged across all sites. The majority of OTUs had a close hit in the BLAST search (97%), although a small number of OTUs (13) did not contain a match in the database (Fig. 2). Taxonomic diversity ranged across the tree of life, with many OTUs containing hits to fungi, and to a lesser extent, various arthropod groups, including insects and mites. In addition, numerous other groups were recovered in the searches, including protozoans, nematodes, an annelid and even a vertebrate (*Sus scrofa*, wild boar). Rarefaction curves for each sample site suggest that OTU diversity has not yet been reached, indicating that the pitcher plant community was not fully sampled in any of the sites (Fig. 3). Although fewer sequences per site likely prevented us from obtaining representatives from the full diversity of species within each pitcher, wider spatial sampling helped us achieve our goal of sampling a large number of eukaryotic species for a comparative data set (see below).

## Population structure

A global clustering effort was completed to generate a comparative data set for taxa that span the Mississippi River. As we were interested in widespread taxa, we used all sequences collected from Lake Ramsey—including those collected from additional time periods—resulting in the use of the full data set (8,991 sequences). Following *de novo* clustering, UPARSE produced 323 OTUs of which 65 contained a minimum of ten sequences and of these, 31 OTUs contained at least three representatives on either side of the river. BLAST hits of a single sequence from each of the 31 OTUs indicate that fungi and mites are the most well represented taxa (Table 1). One OTU did not contain a significant BLAST hit, and with parameters relaxed, poorly matched a portion of the sequence to multiple disparate taxonomic groups. Since we detected it in multiple pitchers, it seems unlikely that this OTU represents a chimeric sequence. Given the incompleteness of taxonomic databases, however, we retained this OTU for downstream analysis, resulting in a final dataset of 31 OTUs (see Table S1 for the sequencing distribution among locales). In this final set, the number of sequences per OTU ranged from 14 to 2,507, with a median of 54 (average of 225 sequences; Table 1).

A range of genetic variation is present in the sampled OTUs (Table 1). For example, estimates of nucleotide diversity ($\pi$) range from ~0.001 to 0.05, a fifty fold difference. Tajima $D$ values are negative for most taxa (median $= -2.0101$), with 21 of these values significant, indicative of an excessive number of segregating sites in the data sets. Negative Tajima $D$ values can be interpreted as resulting from a rapid demographic expansion, or

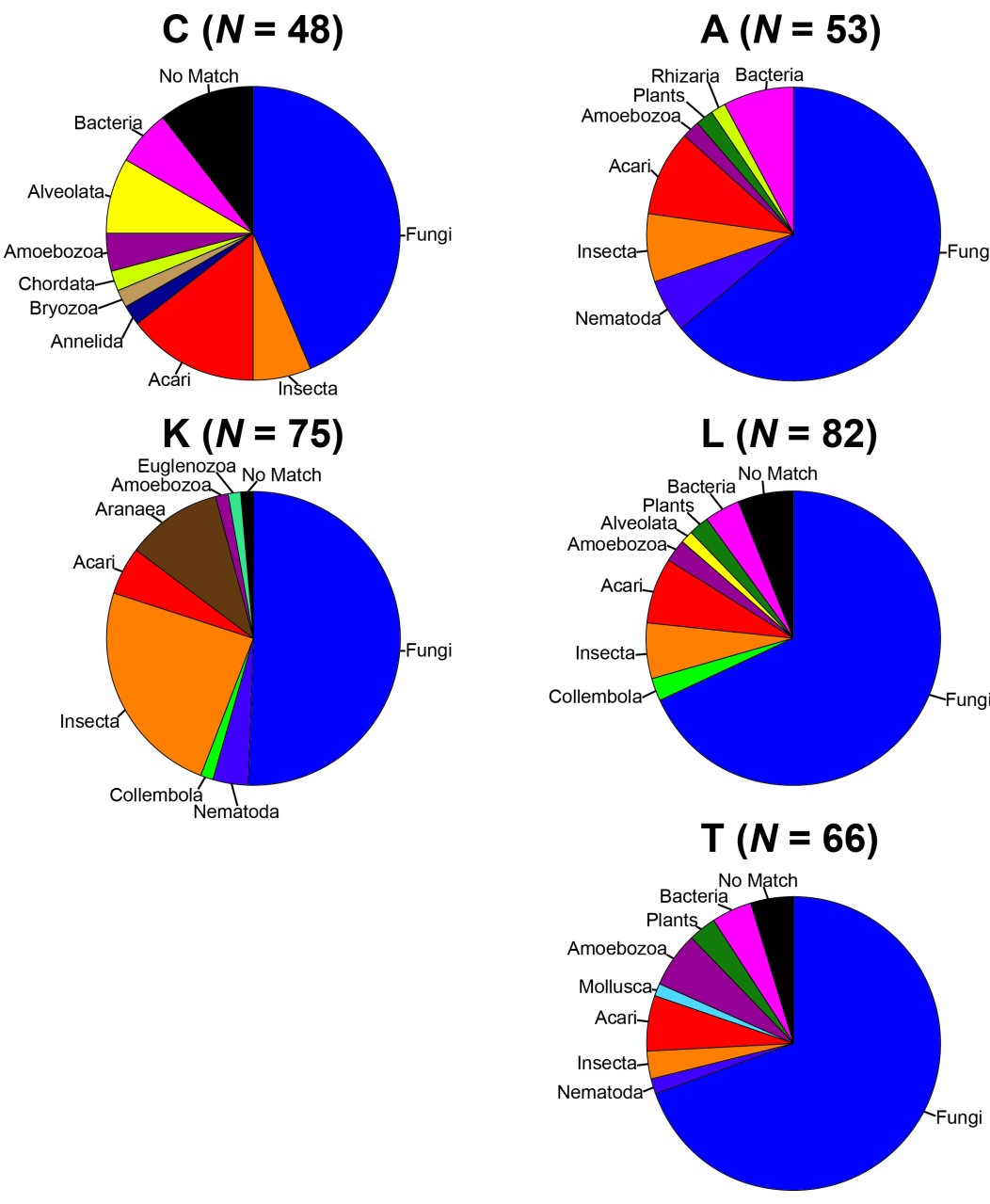

**Figure 2** **Taxonomic composition of the OTUs for each sample site.** Each site contains the number of OTUs (*N*) and the major lineages in which they belong. See Supplemental Information 1 for full sampling information.

from natural selection, on the marker itself or on a linked gene. This could also be the result of population structure in those OTUs, as collapsing separate populations can increase the number of segregating sites in a taxon. Among taxonomic groups, all fungi have a negative Tajima *D* value, with the majority (73%) being significant. Of note are the Tajima *D* values for the arthropods, where all three insects have significantly negative values, and all seven mites have negative values, with three out of seven being significant.

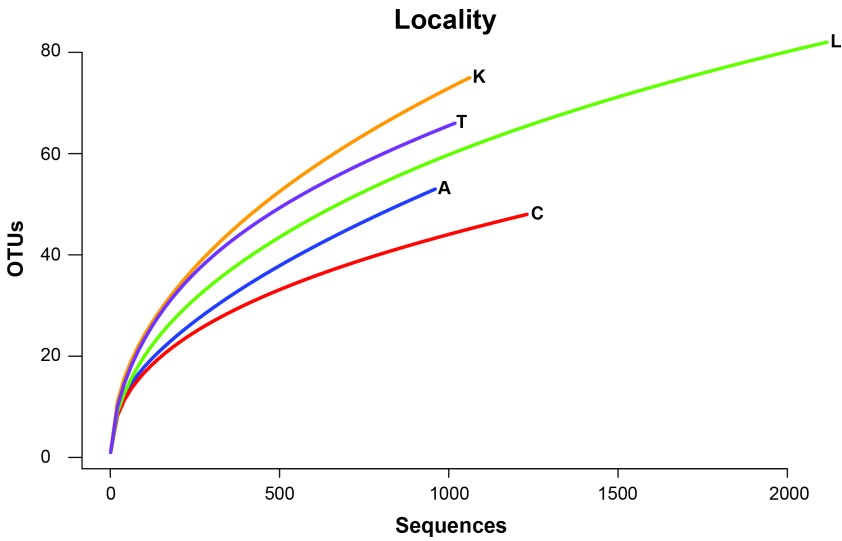

**Figure 3** Rarefaction curves of OTU richness at each sampling site are shown with the number of OTUs plotted against the number of sequences.

There are varying levels of population structure across the taxonomic groups. Roughly half of the fungi contain significant partitioning of genetic variance at the level of the sampling locale, with two taxa also significant at the level of locales within regions (Fig. 4; Table S2). Sequence-based F statistics display similar patterns, with $G_{ST}$ values ranging from 0.003 to 0.280 (average $G_{ST} = 0.101$), suggesting population genetic structure is evident on either side of the Mississippi River in many taxa. Despite this structure, there is considerable sharing of alleles across the Mississippi River in the fungi, although some of the species contain greater sorting than would be expected by chance (see GSI results; Fig. 4; Table S2). Furthermore, genetic diversity in all but two of the fungi is not correlated with geographic distance (Table S2). Results in the mites are similar, with roughly half of the taxa sampled showing a significant amount of genetic variation distributed among the locales, as well as across the Mississippi River (Fig. 4, Table S2). F statistics in the mites are slightly lower than those in the fungi (average $G_{ST} = 0.081$). This structure is also evident in the GSI results, with allelic sorting in most taxa higher than would be expected by chance (Fig. 4; Table S2). Patterns among the fungi (roughly half of the OTUs), mites and insects generally reflect those of the host plant, with the remaining taxa showing essentially no evidence for this shared genetic structure. Chi-squared goodness-of-fit tests show that more taxa share population genetic structure with the host plant than would be expected by chance in many of the analyses (Table 2).

## DISCUSSION

Investigations into the evolutionary history of host plants and their associated insects have provided evidence for co-diversification over long time-periods (e.g., *Weiblen & Bush, 2002*) in addition to demographic patterns that suggest a concerted response to abiotic factors over shorter periods of time (e.g., *Smith et al., 2011*). Inspired by such studies, we

**Table 1 Taxa included in the final comparative data set.** Information for OTUs include number of sequences ($N$), their nearest BLAST hit (except for *S. alata*), nucleotide diversity ($\pi$), Watterson's theta ($\Theta_w$) per site, Tajima's D, and $G_{ST}$. Significance of $G_{ST}$ and Tajima's D (D following a beta distribution; *Tajima, 1989*) at $\alpha = 0.05$ is indicated with an asterisk ($\star$).

| Taxa | $N$ | BLAST | $\pi$ | $\Theta_w$ | Tajima's $D$ | $G_{ST}$ |
|------|-----|-------|-------|------------|--------------|----------|
| Fungi1 | 52 | *Cladosporium sp.* (Fungi) | 0.0088 | 0.0244 | −2.1585$\star$ | 0.1099$\star$ |
| Fungi2 | 51 | *Fusarium annulatum* (Fungi) | 0.0130 | 0.0299 | −2.0211$\star$ | 0.2335 |
| Fungi3 | 22 | *Curvularia sp.* (Fungi) | 0.0157 | 0.0272 | −1.7148 | 0.1419 |
| Fungi4 | 2,507 | *Candida saitoana* (Fungi) | 0.0059 | 0.0918 | −2.6902$\star$ | 0.0025 |
| Fungi5 | 97 | *Candida saitoana* (Fungi) | 0.0072 | 0.0203 | −2.0520$\star$ | 0.0514 |
| Fungi6 | 168 | *Candida saitoana* (Fungi) | 0.0090 | 0.0187 | −1.6674 | 0.0167 |
| Fungi7 | 84 | *Candida saitoana* (Fungi) | 0.0088 | 0.0217 | −2.0016$\star$ | 0.0711 |
| Fungi8 | 57 | *Candida quercitrusa* (Fungi) | 0.0134 | 0.0341 | −2.0901$\star$ | 0.1150 |
| Fungi9 | 30 | *Candida saitoana* (Fungi) | 0.0010 | 0.0053 | −2.3512$\star$ | 0.0991 |
| Fungi10 | 54 | *Candida saitoana* (Fungi) | 0.0050 | 0.0183 | −2.3958$\star$ | 0.0788 |
| Fungi11 | 189 | *Mucor circinelloides* (Fungi) | 0.0078 | 0.0307 | −2.3694$\star$ | 0.0090 |
| Fungi12 | 14 | Uncultured soil fungus (Fungi) | 0.0087 | 0.0100 | −0.6488 | 0.2805 |
| Fungi13 | 40 | Uncultured fungus (Fungi) | 0.0103 | 0.0125 | −0.6810 | 0.2494 |
| Fungi14 | 766 | Fungal endophyte (Fungi) | 0.0117 | 0.0586 | −2.4374$\star$ | 0.0052 |
| Fungi15 | 15 | *Nigrospora sphaerica* (Fungi) | 0.0106 | 0.0188 | −1.8028$\star$ | 0.0451 |
| Amoebozoa1 | 227 | *Fuligo septica* (Amoebozoa) | 0.0042 | 0.0347 | −2.6856$\star$ | 0.0411 |
| Alveolata1 | 18 | *Leptopharynx costatus* (Alveolata) | 0.0182 | 0.0324 | −1.8094$\star$ | 0.0418 |
| Nematoda1 | 79 | *Nematoda sp.* (Nematoda) | 0.0150 | 0.0317 | −1.8654$\star$ | 0.4101 |
| Nematoda2 | 154 | *Nematoda sp.* (Nematoda) | 0.0130 | 0.0174 | −0.9456 | 0.2359 |
| Nematoda3 | 21 | *Nematoda sp.* (Nematoda) | 0.0188 | 0.0170 | 0.2858 | 0.0624 |
| Insect1 | 61 | *Brachymyrmex depilis* (Insecta) | 0.0122 | 0.0285 | −2.0101$\star$ | 0.0346 |
| Insect2 | 37 | *Solenopsis xyloni* (Insecta) | 0.0208 | 0.0409 | −1.8296$\star$ | 0.0723 |
| Insect3 | 41 | *Paratrechina hystrix* (Insecta) | 0.0081 | 0.0208 | −2.1225$\star$ | 0.2694 |
| Mite1 | 828 | *Ovanoetus sp.* (Acari) | 0.0086 | 0.0620 | −2.5727$\star$ | 0.0150$\star$ |
| Mite2 | 30 | *Ovanoetus sp.* (Acari) | 0.0152 | 0.0348 | −2.1831$\star$ | 0.2911 |
| Mite3 | 1,071 | *Anoetus sp.* (Acari) | 0.0071 | 0.0678 | −2.6276$\star$ | 0.0101$\star$ |
| Mite4 | 56 | *Anoetus sp.* (Acari) | 0.0114 | 0.0242 | −1.7765 | 0.0437 |
| Mite5 | 34 | *Anoetus sp.* (Acari) | 0.0176 | 0.0219 | −0.7551 | 0.1516 |
| Mite6 | 50 | *Anoetus sp.* (Acari) | 0.0111 | 0.0197 | −1.4951 | 0.0427 |
| Mite7 | 45 | *Anoetus sp.* (Acari) | 0.0049 | 0.0098 | −1.5594 | 0.0147 |
| Unknown | 66 | No BLAST Match | 0.0059 | 0.0198 | −2.2971$\star$ | 0.3539 |
| Host plant | 79 | *Sarracenia alata* | 0.0028 | 0.0034 | −0.4521 | 0.8483$\star$ |

**Table 2 A chi-squared goodness-of-fit test was used to measure if the number of taxa with significant population genetic structure was more than would be expected by chance alone.** Under a null model we would expect a significant result 5% of the time (assuming $\alpha = 0.05$). Results show that for many analyses, there are more OTUs with significant values than expected by chance, suggesting an association between many members of the community and the host pitcher plant.

| Test | $\chi^2$ | df | $p$-value | Number significant | Total taxa |
|------|----------|-----|-----------|--------------------|------------|
| $\Phi_{SC}$ | 22.3612 | 1 | $2.26 \times 10^{-6}$ | 7 | 29 |
| $\Phi_{ST}$ | 80.8004 | 1 | $2.20 \times 10^{-16}$ | 12 | 29 |
| $\Phi_{CT}$ | 1.5263 | 1 | 0.22 | 0 | 29 |
| $G_{ST}$ | 1.4278 | 1 | 0.23 | 3 | 31 |
| $GSI_E$ | 20.1715 | 1 | $7.08 \times 10^{-6}$ | 7 | 31 |
| $GSI_W$ | 13.4482 | 1 | $2.45 \times 10^{-4}$ | 6 | 31 |

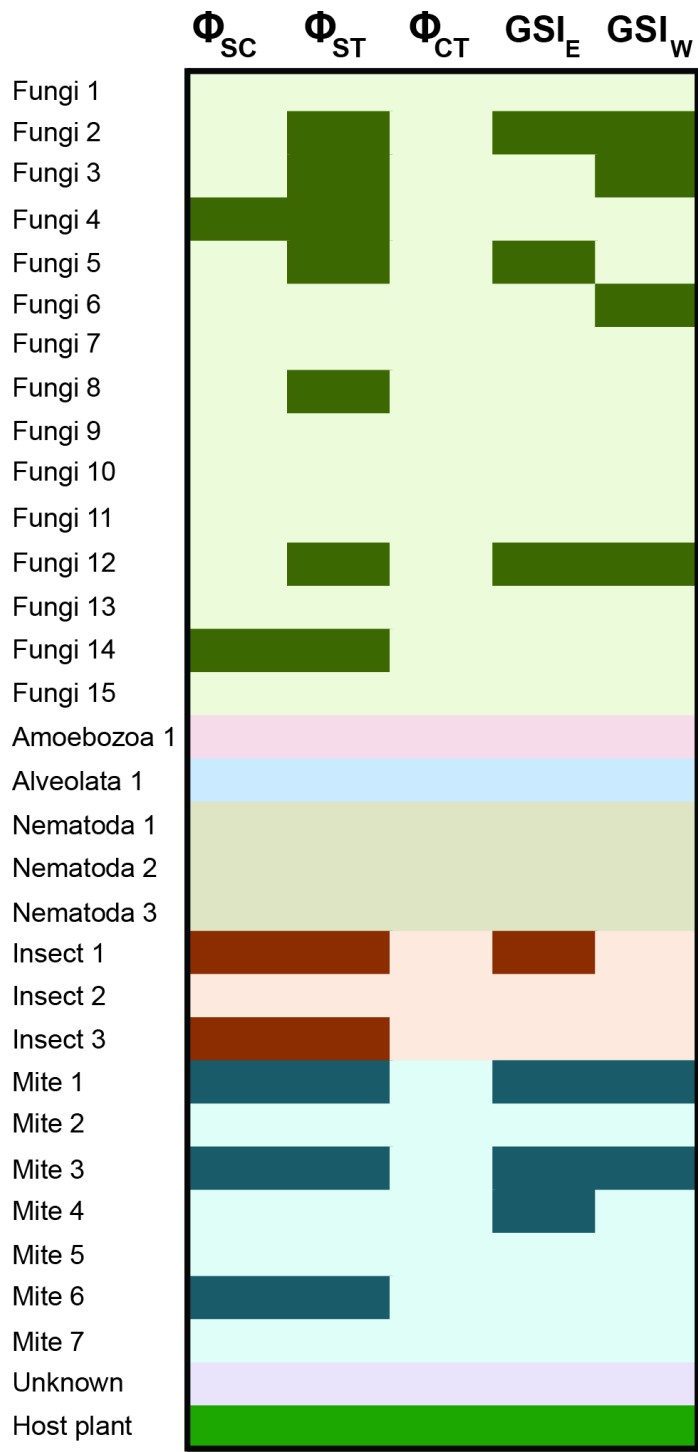

**Figure 4** **Population genetic structure for the inquiline community spanning the Mississippi River.** Results are shown from the AMOVA and GSI analyses. AMOVA analyses show the hierarchical partitioning scheme of locales within regions ($\Phi_{SC}$), locales within total distribution ($\Phi_{ST}$), and between regions ($\Phi_{CT}$). GSI analyses represent the amount of allelic sorting on the eastern and western sides of the Mississippi River. Dark cells indicate taxa with significant genetic structure at the corresponding level; Table S2 contains specific values from each analysis. See *Carstens & Satler (2013)* for sampling information for *S. alata*, as these samples were collected from throughout the plant's distribution.

sampled a diverse set of organisms (representing similarly diverse ecological interactions) associated with the Pale Pitcher Plant in order to investigate the extent to which this ecological community has co-diversified. Within *Sarracenia alata*, previous work has demonstrated that populations are genetically structured across the landscape (*Koopman & Carstens, 2010*). Major rivers are important drivers of diversification both in the region (*Soltis et al., 2006*) and for *S. alata* (*Zellmer et al., 2012*), and analysis of the barcode data presented here demonstrates that slightly under half of the eukaryotes sampled share similar population genetic structure with *S. alata*.

Our results show that a core eukaryotic community exhibits congruent patterns of population genetic structure, with many taxa displaying significant genetic structuring at the level of the sampling locality (based on $\Phi_{ST}$); approximately half of the microscopic fungi and half of the mites are structured in a manner similar to that of the host plant (Fig. 4). Given the dispersal capabilities of fungal spores (e.g., *Peay, Kennedy & Bruns, 2008*), this degree of population genetic structure is strikingly high (but see *Taylor et al., 2006*). Fungi are ubiquitous in terrestrial habitats, however, with many species associated with soils and plants. Fungal species have also been recovered from pitcher plant leaves, demonstrating their known presence within these microhabitats (reviewed in *Adlassnig, Peroutka & Lendl, 2011*). Multiple mite species from the family Histiostomatidae have been described from *Sarracenia* pitcher plants (*Hunter & Hunter, 1964*; *Fashing & O'Connor, 1984*), and act as prey consumers within the pitchers. Approximately half of the mites identified here exhibited population genetic structure similar to that of *S. alata*, reflecting structure seen both among sample sites and across the Mississippi River (Fig. 4). Other members of this core group include two of the three sampled insects (all three share a closest BLAST hit to ants), with general strong support across analyses for co-diversification. Ants comprise a large component of prey items for *Sarracenia* (*Newell & Nastase, 1998*; *Ellison & Gotelli, 2009*), and one could interpret these results as being indicative of general biogeographic structuring in the longleaf pine savannah habitat. Such results are illustrative of the challenges associated with understanding if interactions among organisms have driven the shared responses to historical events reflected in patterns of co-diversification.

Although ants represent a major prey item of *Sarracenia* plants, our field work suggests that the plant does not specialize on any particular species of ant. Therefore, ants in general may be considered to have a relatively weak ecological association with the plant, and these results may highlight the strong influence that landscape and abiotic factors have on diversification in the region. Teasing these two interpretations (i.e., ecological association vs. landscape and abiotic factors) apart is non-trivial, yet an understanding of the strength of ecological association, habitat affinity, and dispersal ability can lend insight on this issue. Given the ecological associations shared among many of the inquilines with the host plant in this system, shared population structure does provide evidence that ecology plays a role in shaping diversification patterns through time. As the Mississippi River is an important evolutionary barrier to this system (and many groups across the region), diversification across the river may have taken place via the mechanism of oxbow lake formation, where changes in the river channel moved a portion of the habitat from the eastern to the western side of this barrier.

While slightly under one-half of the sampled taxa share population genetic structure with *S. alata*, there are other taxa with discordant evolutionary histories. Many of the fungal taxa exhibit little to no population structure, and we suspect that these microscopic species are widespread and not restricted to the pitcher plant bog habitats. Their dispersal ability is likely to be higher than the larger members of this community, allowing them to escape the influence of biogeographic barriers. Other microscopic eukaryotes exhibit no evidence of population structure, including two protizoans and the sampled nematodes, suggesting that biogeographic barriers do not provide an obstacle for long-distance dispersal in these taxa (*Finlay, 2002*). In addition, one insect species demonstrates a lack of population structure. Further investigation of the BLAST result for this OTU (hit to *Solenopsis xyloni* in original search; see Table 1) indicates that this OTU is an identical match to the invasive red fire ant (*Solenopsis invicta*). Given the devastating impact and colonization power of the red imported fire ant, the lack of population structure is likely a product of their recent introduction to the southeastern United States (from South America). *Solenopsis invicta* has grown explosively and displaced native species in the region (*Porter & Savignano, 1990*; *Stuble, Kirkman & Carroll, 2009*) and the lack of structure recovered is consistent with the expectations of an invasive species. Clearly, ecological association and dispersal ability both play a role in the level of congruence detected in population genetic structure across species, although quantifying these two factors, especially dispersal for microscopic eukaryotes, remains a challenge.

Phylogeographic patterns within a species can be informative, but in aggregate, the results across many species make it possible to identify community responses to landscape changes. To date, phylogeographic researchers have not fully utilized metagenomics as a tool for increasing the taxonomic breadth of a comparative study. The *S. alata* system is ideal for such studies, as each pitcher provides a self-contained and discreet habitat, where micro- and macroscopic organisms can live and persist in an ecological entanglement. The increased sampling facilitated by metagenomic approaches allowed us to identify a core evolutionary community within *S. alata*, and the simplest explanation for this congruence is that the core community has diversified in unison because the constituent members are ecologically dependent on *S. alata*. As such, the OTUs sampled represent an example of shared evolutionary patterns across an ecological community, and suggests that co-diversification is not limited to specialized interactions such as plants and pollinators. The recent discovery of cryptic diversity within *S. alata* (*Carstens & Satler, 2013*), together with the work presented here, highlights the need for conserving species like pitcher plants, which play a role in the survival of many different organisms. Such systems contain species that have been ecologically interdependent over evolutionary time scales, thus the loss of substantial diversity in the pitcher plant could lead to loss of diversity in its commensal species.

Given the power of comparative analysis for phylogeographic research, metagenomics can be leveraged to increase our knowledge of the evolutionary processes that lead to biogeographic patterns around the world. In particular, environmental sampling can provide access to taxa spanning the range of ecological and life history traits, as well as greater spatial sampling, which can provide evidence of the landscape processes that have

structured species and communities in a region (*Bermingham & Moritz, 1998*). Potential pitfalls, however, do remain when applying metagenomics for such an analysis. In this study, a large number of sampled OTUs are fungi (Fig. 2), which could be indicative of their ubiquity in nature, but could also be due to our use of primers originally developed from fungal genomic resources. The need to isolate specific gene fragments with primers could have biased the taxonomic sampling, which may have also contributed to the non-asymptotic nature of the rarefaction curves, although this is more likely due to a relatively small number of sequences from next generation sequencing with the sampling strategy used in this study. In addition, challenges exist when using *de novo* assembly for generating a taxonomic data set, particularly with the requirement of a percent threshold to determine the placement of sequences within OTUs. Although some values are commonly used for certain groups, it is unlikely that a single cutoff is appropriate across the tree of life. Further exploration of the correlation between sequence similarity and taxonomic identity across diverse groups is necessary to better place sequences with their proper OTU. However, as demonstrated here, metagenomic data can be beneficial for phylogeographic studies, with careful and transparent analysis of the data providing valuable insight into the diversification of a region, or in our case, an ecological community composed of a diverse set of lineages.

Remarkably, the co-diversification described here may extend beyond the eukaryotic members of this ecosystem. *Koopman & Carstens (2011)* provide evidence that the phylogenetic community structure in the bacterial microbiome reflects the population genetic structure of the plant. Since the bacterial microbiome is dominated by Enterobacteriaceae (*Koopman et al., 2010*), a family commonly found in animal guts, it could be that the insect members of the core community facilitate colonization of bacteria in the pitchers (which are sterile before opening; see *Peterson et al., 2008*). If the core arthropods seed the pitchers with Enterobacteria, these microbes may produce enzymes that contribute to the digestive function of the pitcher. Since these complex ecological interactions have likely persisted for hundreds of thousands of years (based on estimates from *S. alata*), our work underscores the importance of investigating the evolutionary relationships of ecological communities.

## ACKNOWLEDGEMENTS

We thank members of the Carstens lab, particularly Margaret Hanes (formerly Koopman) and Sarah Hird, for comments and discussion regarding the manuscript. We thank the Academic Editor and reviewers for comments that helped improve this manuscript.

### Funding

Funding was provided by the Society for the Study of Evolution (Rosemary Grant Award for Graduate Student Research) and the Society of Systematic Biologists (Graduate Student Research Award) to JDS; and the Louisiana EPSCoR Pfund, the LSU Office of Research

and Economic Development, and Louisiana Board of Regents for funding to BCC. The funders had no role in study design, data collection and analysis, decision to publish, or preparation of the manuscript.

## Grant Disclosures

The following grant information was disclosed by the authors:
Society for the Study of Evolution.
Society of Systematic Biologists.
Louisiana EPSCoR Pfund.
LSU Office of Research and Economic Development.
Louisiana Board of Regents.

## Competing Interests

The authors declare there are no competing interests.

## Author Contributions

- Jordan D. Satler conceived and designed the experiments, performed the experiments, analyzed the data, contributed reagents/materials/analysis tools, wrote the paper, prepared figures and/or tables, reviewed drafts of the paper.
- Amanda J. Zellmer conceived and designed the experiments, performed the experiments, analyzed the data, contributed reagents/materials/analysis tools, wrote the paper, reviewed drafts of the paper, collected the samples.
- Bryan C. Carstens conceived and designed the experiments, contributed reagents/materials/analysis tools, wrote the paper, reviewed drafts of the paper, collected the samples, purchased coffee.

## Data Availability

Dryad doi: 10.5061/dryad.j3n0g.

## Supplemental Information

Supplemental information for this article can be found online at http://dx.doi.org/10.7717/peerj.1576#supplemental-information.

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
