# Peer review of "Biogeographic barriers drive co-diversification within associated eukaryotes of the Sarracenia alata pitcher plant system"

_PeerJ, doi:10.7717/peerj.1576_

## Round 0.1 · original submission · Minor Revisions

The major issues raised by Reviewer 1 will likely require the most work to address. What can be inferred regarding co-diversification without a quantitative comparison of differentiation relative to the host plant and (more challengingly) relative to the external biota? I might suggest at least adding the results for the host plant to Figure 4, assuming they are already at hand.

Some of the minor comments are also worth pointing out for special attention. I do agree with Reviewer 3 that the use of the term 'metagenomics' for environmental sequencing of a 275bp locus is misleading. I am also not clear why the analysis ignores taxa with less than 3 representatives on one side of the river; doesn't this just downwardly bias estimates of structure at the regional scale, and unnecessarily discard evidence for differentiation?

In my own reading, I had a number of additional comments and questions that I would ask the authors to address.

1. Why not exclude the April-May samples from Lake Ramsay, or at least segregate in the analyses, in order to have comparable diversity measurements across sites? See also the comment on sampling times from Reviewer 1.

2. There is no surprise at seeing a negative Tajima's D when pooling data across populations, because allele frequency differentiation results in an excess of segregating sites. So, significance tests for D=0 do not say much; these values are best interpreted relative to D in the host plant. I believe this can be addressed together with the response to Reviewer 1's first major issue.

3. I am unclear as to why the allelic sorting analysis is applied to populations on either side of the Mississippi separately. Please clarify the rationale and/or show values for E & W combined.

4. Lack of significance in phi_CT seems counter to idea that a substantial amount of biogeographic differentiation is explained by the Mississippi River Instead, it seems that most of the differentiation is between populations instead. Is that not contrary to the conclusion on line 358?

5. Please be sure to also include supplemental Table S2 in the next submission, so that it is possible to inspect the quantitative results that are shown schematically (at only one significance threshold) in Fig 4.

Reviewer 1 ·

Basic reporting

This manuscript addresses whether pitcher plants and their associated inquiline community have co-diversified across their range. Although not explicitly addressed in the introduction, this research is closely related to the field of community genetics, in which each genotype of host plant may have an extended phenotype consisting of an associated community. The topic is an important one, as we are increasingly finding that genetic diversity in one species plays a large role in determining the dynamics of other associated species. The authors here are using novel molecular approaches to address this question, as they use genomics to examine all eukaryotes within pitcher plants.

Experimental design

I am satisfied with the experimental design.

Validity of the findings

Unfortunately, I do not believe that the data necessarily support the authors’ conclusions. This study would benefit greatly from the use of a null model to determine whether the observed patterns exceed what we would expect by chance alone. The evidence presented here suggests that about half of the eukaryotic species show evidence of diversification with the plant. Is 50% more than we would expect? Certainly we don’t expect zero association. If both the plants and the eukaryotes were distributed randomly, we would expect some areas of overlap between them. So what level then is considered significantly greater than random? A null model based on the distribution of both plants and the eukaryote community would help to determine this.

It would also be helpful to know what the eukaryotic community is like in the neighborhood outside of the plants. Are these eukaryotes specialists in pitcher plants (probably true of some of them) or can they also be found in other habitats? If the latter, then eukaryotes and plants could have diversified in response to similar habitat effects, but not necessarily in response to one another.

Additional comments:

Line 33: what is the evidence that the inquiline community is mutualistic? There is some evidence that bacteria are beneficial to pitcher plants, but if so, then consumers of bacteria are likely to be parasitic.

Line 75: I would argue that pollinators and plants are mutualists, but not symbiotic, as pollinators spend most of their time unassociated with the plant.

Line 94: Again, I would argue that this statement applies to the bacteria, but not necessarily the eukaryotes.

Line 164: The 28S subunit is only present in eukaryotes, so I am surprised to see bacteria listed as a taxonomic group in Fig. 2. As such, the figure is misleading, not only because of the presence of bacteria, but also because if one DID sequence the bacteria, their species diversity would likely greatly outnumber the eukaryotes in the community.

Line 318-320: Please explain what these values mean biologically.

·

Basic reporting

This article is written very clearly; the relevant literature is cited and introduction & background explains the particular importance of this study; Figures are appropriate;

Experimental design

The experimental design is planed and carried out very carefully and accurate.

Validity of the findings

This study is very innovative and represents a framework in co-evolution of eukaryotic inquilines in phytotelms of carnivorous pitcher plants. I highly recommend the acceptance of this article.

Additional comments

I highly recommend the acceptance of this article. I only have one formal suggestion and one question:
Line 397: to avoid confusions with the genus Sarracenia, the genus Solenopsis should be fully announced if mentioned for the first time in the text
Line 400: S. invicta and S. invictus (female/male form) are these scientific names used synonymous?

·

Basic reporting

The article is exceptionally well written with a sufficient introduction and background section that appropriately cites the relevant literature.

Experimental design

The experimental design is original and well thought through. The research question is clearly identified and the methods of analysis of the eukaryotic 28S data are appropriate; I cannot comment on the methods of analysis used for assessing genetic variation in the host as these are largely outside of my area of expertise.

A few comments:

Line 162: It would be helpful if the authors listed sampling times for the five locations, as they have indicated that sampling occured in June and again August. As the summer months in the South East are typically accompanied by regular precipitation, one might expect that if sampling on either side of the Mississippi occurred in different months, there may be an environmental affect that has influenced community data structure. I doubt this is a serious issue, but it would be useful to have it addressed.

Validity of the findings

The findings are presented clearly; the only criticism of the manuscript in this section is the depth of sequencing achieved, which is considerably lower than is standard for research in the field and reduces the level of statistical significance that can be assigned to comparisons between the community data.

Additional comments

I thoroughly enjoyed this research article, and have only minor comments.

Line 37: The term metagenomics is more commonly reserved for cases in which shotgun metagenomics have been employed. I would alter this statement to state that you employed 454 amplicon-based metagenomics.

Line 163: "DNA was extracted", rather than "Fluid was extracted"

Line 177: It is unclear what is meant by the removal of redundant sequences; are the authors referring to dereplication at 100% identity? If so, I would rephrase as these sequences are not removed but rather compressed.

Line 180: I have not heard this term used to describe the methods used in UPARSE; I would suggest that authors check to make certain this is in fact the terminology they intend to use. It is also sufficient to state that "sequences were clustered into operational taxonomic units using UPARSE".

Line 185: I am a little unclear on the clustering approach; my interpretation is that sequences were clustered in two ways: 1) separately for each location first, and 2) combined for all locations. It could be helpful for the authors to put a statement earlier in the methods indicating their use of two different approaches to analyzing the data.

Line 228: It is unclear why the authors chose to ignore OTUs that may be specific to one side of the Mississippi, as they are trying to test the hypothesis that the position with respect to the river is a major determinant of community structure. Perhaps there were not many OTUs that were unique to one side of the river but were found in more than one sample or location, which could be spurious sequence artifacts.

Line 361: See also Taylor, 2006 in Biological sciences, which speaks to issues of dispersal limitation in fungi.

Line 422: grammar

---

## Round 0.2 · accepted · Accept

I appreciate the detailed response to the reviewers' comments and conclude that the revisions merit acceptance of the manuscript.

Reviewer 1 ·

Basic reporting

I reviewed this manuscript previously and I am satisfied with the authors' changes in response to my previous comments.

Experimental design

I reviewed this manuscript previously and I am satisfied with the authors' changes in response to my previous comments.

Validity of the findings

I reviewed this manuscript previously and I am satisfied with the authors' changes in response to my previous comments.

·

Basic reporting

The basic reporting is sufficient for publication.

Experimental design

The experimental design is sufficient for publication.

Validity of the findings

The validity of the findings is sufficient for publication.

Additional comments

The revisions made by the author have addressed my minor concerns, mainly by clarifying methods and assumptions made in their analyses. I endorse publication of the manuscript.